# Galectins in Intra- and Extracellular Vesicles

**DOI:** 10.3390/biom10091232

**Published:** 2020-08-24

**Authors:** Sebastian Bänfer, Ralf Jacob

**Affiliations:** 1Department of Cell Biology and Cell Pathology, Philipps-Universität Marburg, 35043 Marburg, Germany; baenfers@staff.uni-marburg.de; 2DFG Research Training Group, Membrane Plasticity in Tissue Development and Remodeling, GRK 2213, Philipps University of Marburg, 35043 Marburg, Germany

**Keywords:** galectin, non-classical secretion, endosome, exosome, sorting

## Abstract

Carbohydrate-binding galectins are expressed in various tissues of multicellular organisms. They are involved in autophagy, cell migration, immune response, inflammation, intracellular transport, and signaling. In recent years, novel roles of galectin-interaction with membrane components have been characterized, which lead to the formation of vesicles with diverse functions. These vesicles are part of intracellular transport pathways, belong to the cellular degradation machinery, or can be released for cell-to-cell communication. Several characteristics of galectins in the lumen or at the membrane of newly formed vesicular structures are discussed in this review and illustrate the need to fully elucidate their contributions at the molecular and structural level.

## 1. Introduction

Vesicle-mediated traffic and communication is crucial for all tissue and organ function. These membrane-covered transport containers can passage cargo between intracellular compartments and between cells or tissues if released into the extracellular milieu. They are composed of various biomolecules including carbohydrates, lipids, nucleic acids, and proteins, a composition that is ultimately linked to their function. Synaptic vesicles, for example, are dominated by proteins essential for membrane traffic and neurotransmitter uptake [1]. They are replenished by high-fidelity sorting machinery following fusion with the presynaptic membrane [2]. Specific protein sorting into such clearly defined vesicle populations depends on short amino acid motifs, which are bound by corresponding receptors, or might involve co- or posttranslational modifications of individual cargo polypeptides. Glycans are prominent examples for protein modifications that direct glycoproteins into lysosomal organelles [3], to the apical membrane domain of polarized epithelial cells [4,5,6] or that target unsalvageable glycoproteins for endoplasmic reticulum-associated degradation [7]. These glycan-dependent sorting steps are mediated by sugar-binding lectins, which bind, tag or cluster glycosylated target molecules for further processing.

## 2. The Galectin Family

A family of soluble, non-glycosylated lectins of small molecular weight between 14 to 39 kDa is the galectin family. These lectins have affinities for β-galactoside structures, in combination with a consensus core sequence, located in the carbohydrate recognition domain (CRD) [8]. The galectinic CRD consists of ~130 amino acids, which build a concave groove that defines the galectin-carbohydrate recognition site [9]. Based on the number and organization of their CRDs, galectins have been classified into the “prototypic” galectins with only one CRD, the “chimeric” type with one CRD and an N-terminal domain and the “tandem-repeat” type with two homologous but not identical CRDs [10]. Galectins have been highly conserved during evolution. However, no known homolog has yet been described for unicellular organisms, suggesting that these lectins appeared concomitantly with the development of multicellular organisms [11]. Their involvement has been described for sugar-dependent and -independent modulations of multiple cellular functions such as the cell cycle, apoptosis, RNA transcription, cell differentiation, and also in the organism development or the progression of pathological conditions, such as cancerogenesis, immune response, or even pathogen entry [12,13]. Galectins do not contain an N-terminal signaling sequence to be exported by the cell. Their place of synthesis is the cytoplasm, from where they are directed to various cell regions. Some of these regions, which are linked to vesicular trafficking pathways, are discussed within this review.

## 3. Galectins in Intracellular Vesicular Compartments

Distinct members of the galectin family can monitor membrane rupture of endosomes or lysosomes through binding of host β-galactose-containing glycoconjugates exposed in the cytoplasm after membrane damage. Galectin-3 has been found in the vicinity of invading bacteria after lysis of their phagocytic vacuole [14,15]. The idea is that host cells employ a common mechanism that targets invading pathogens for selective autophagy (Figure 1). Galectin-3 then plays the role of spotting intracellular locations of vacuole lysis. Once lysosomal rupture has been marked, autophagosomes selectively sequester the damaged lysosomes. In this process, a membrane engulfs the lysosomes and closure of this membrane results in the formation of a double-membrane autophagosome. It has also been demonstrated that lysosomal damage caused by certain lysosomotropic agents was characterized by galectin-3 recruitment [16]. In line with these observations, galectin-3 was recruited to damaged endosomes containing internalized calcium phosphate precipitates [17]. These endosomes were found to then colocalize with LC3-positive autophagosomes. The lectin thus marks damaged endosomes for autophagy in a manner dependent on the autophagy adapter protein p62, which has been shown to recruit autophagosomal components [18]. Data from this study also indicated that other galectins may be found on endocytosed damaged vesicles. Indeed, Thurston et al. have found that galectin-8 is recruited to damaged phagosomes containing *Salmonella* in a similar fashion [19]. Galectin-8 activates antibacterial autophagy through association with the autophagy machinery. Dependent on the autophagy adapter protein, NDP52, this results in autophagy of the damaged bacteria-containing vesicles to eliminate bacterial invasion. Galectin-mediated labelling of damaged endosomes moreover seems to rely on the composition of carbohydrates recognized by the lectins. Structured illumination microscopy revealed that galectin-3 and galectin-8 localize in distinct microdomains of damaged endosomes; galectin-3 mainly distributes in the outer region, while galectin-8 primarily accumulates in the inner region around damaged endosomes [20]. This study also suggests that alterations in the composition of surface glycans determine the role each lectin plays in autophagic activation. On the other hand, galectin-8 but not galectin-3 is involved in secretory autophagy of the cytosolic cargo IL-1β [21]. The process is mediated by complex formation of galectin-8 with the tripartite motif-containing cargo receptor TRIM16 and other protein complexes. These act in a sequential manner to respond to triggers, and to initiate and execute unconventional secretion of cargo. TRIM16 also interacts with galectin-3 and key autophagy regulators to ensure autophagic protection against lysosomal damage and *Mycobacterium tuberculosis* invasion [22].

Further work has demonstrated that apart from galectin-3 and galectin-8, other galectins, including galectin-1, galectin-4, and galectin-9, are targeted towards impaired endocytic vesicles [23,24]. Remarkably, the timing of galectin-recruitment seems to differ between distinct galectin variants. Galectin-3, galectin-9, and galectin-8 are present at the damaged *C. burnetii* replicative vacuole membrane from early times of infection. In contrast, galectin-1 is targeted at later times of infection, which again indicates that differences in the composition of the carbohydrates exposed at the inner surface of the C. burnetii vacuole lead to specific recognition of distinct galectins. It is thus tempting to conclude that galectins can serve as intracellular sensors that decrypt complex information compiled on membrane surface sugar codes to direct damaged vesicles for autophagy and protect the cell against lysosomal damage [20,25].

Another option how galectins can enter intracellular endocytic and recycling compartments is by endocytosis from the cell surface [26,27,28]. The efficiency of galectin-3 internalization is pH-dependent and can be competitively inhibited by lactose [29]. Moreover, galectin-lattice formation modulates the uptake efficiency of galectins [30]. Interaction between galectin-3, glycosphingolipids, and the cargo proteins β1-integrin or CD44 drive clathrin-independent endocytosis by creating protein-lipid clusters that bend the plasma membrane to form endocytic vesicular carriers [31]. In a similar fashion, internalized galectin-3 interacts in sorting endosomes with newly synthesized glycosylated cargo molecules to form high molecular weight clusters for apical protein sorting in polarized epithelial cells [32,33] (Figure 1). Polypeptides like the carcinoembryonic antigen-related cell adhesion molecule 1, dipeptidylpeptidase IV, lactase-phlorizin hydrolase, the neurotrophin receptor, sodium-proton exchanger 2, and even beta1-integrin are apically sorted by galectin-3 [34,35,36].

Polarized sorting in these cells involves recycling of apically internalized galectin-3 and the passage through Rab11-positive recycling endosomes [37,38]. Disruption of Rab11- and galectin-3 positive compartments by Vps34 deletion resulted in mis-localization to lysosomes of apical recycling endocytic receptors and apical non-recycling solute carriers [39]. A distinct set of lipid raft-associated apical cargo molecules is sorted by galectin-4 into apical transport carriers [27,40]. Apart from glycoprotein binding, the two galectin-4 CRDs interact specifically and with high affinity to a defined pool of glycosphingolipids, i.e., sulfatides with long fatty acid chains, hydroxylated in position-2. This binding to raft constituents cross-links them to form raft clusters, postulated to build up the membrane microdomains that bud off to generate apical transport carriers. Interestingly, this galectin-4 dependent glycolipid/ glycoprotein recruitment and sorting mechanism also seems to play a role in transcytosis of the transferrin receptor from the basolateral to the apical membrane of epithelial cells [41]. Thus, the current idea is that galectin-dependent glycoprotein-clustering segregates these proteins into defined carrier vesicles.

## 4. Galectins in Extracellular Vesicles

Vesicles secreted by a healthy cell into the outer milieu, which are named extracellular vesicles, provide excellent communication platforms between cells or even organs circulating in a living organism. Two vesicle populations differing in size and origin have been described [42]. Intraluminal vesicles of 40–100 nm diameter are formed by inward budding of multivesicular body membranes and released as exosomes at the plasma membrane. Microvesicles on the other hand are shed directly from the plasma membrane and are in the size range of 200 to 1000 nm. Both of these vesicle populations can be specifically isolated, e.g., by differential centrifugation. Early studies on plasma membrane evaginations, which pinch off to form extracellular vesicles suggested that galectin-1 [43] and galectin-3 [44] are released by microvesicle blebbing. Microvesicle loading with these lectins was also supported by microscopic observations showing patches of galectin-3 [45] or galectin-1 [46] in close proximity to the plasma membrane. In contrast, proteomic analysis of exosomes derived from diverse tissues has provided evidence for intracellular galectin-recruitment into intraluminal vesicles. Galectin-3 was found in exosomes from bladder cancer [47], colon cancer [48,49], dendritic [50], macrophages [51], melanoma [52], ovarian cancer [53], stromal [54], and syncytiotrophoblast cells [55]. Moreover, it was detected in exosomes isolated from urine [56,57], the parotic gland [58], and saliva [59].

Apart from the observation that the composition of exosomes displays tissue-associated protein signatures, alterations in exosomal galectin-3 concentrations can serve as biomarker for distinct disease states. This may be the case for the increased exosomal recruitment of galectin-3 observed for distinct cancer cells. Amongst others, an up-regulation of the expression of galectin-3 was demonstrated for carcinomas of the stomach, liver, pancreas, thryroid gland, ovary, and bladder [60]. As a consequence, elevated cytoplasmic galectin-3 levels are then available for intracellular recruitment of the lectin into intraluminal vesicles, which are then released as exosomes. Evidence for a functional role of exosomal cell-to-cell transfer of galectin-3 comes from a study on the communication between B-cell precursor acute lymphoblastic leukemia cells and stromal cells [54]. Fei et al. found that galectin-3 released from stromal cells is internalized by the target cells and transported into the cell nucleus for auto-induction of galectin-3-expression, which protects target cells against drug treatment. In addition, activation of pathological disease mechanisms like macrophage activation and oxidative stress seems to be caused by elevated galectin-3 levels in exosomes isolated from plasma of patients with atherosclerosis [51]. Galectin-3 by itself can also modulate the exosome composition. This was demonstrated by stimulation of HepG2 cells with the carbohydrate recognition domain of galectin-3, which resulted in the upregulation of certain exosomal proteins [61]. The underlying cellular mechanisms of these observations have not been resolved yet and await clarification in future studies.

Galectin-1 attached to the surface of exosomes from placenta mesenchymal stromal cells is involved in exosome-adhesion, thus aiding in their neuroprotective function [62]. This lectin was further identified on exosomes from tumor and syncytiotrophoblast cells [55,63]. The presence of galectin-5 on the surface of reticulocyte exosomes negatively affects their uptake by macrophages in a lactose-dependent process [64]. This suggests that galectin-5 inhibits exosome uptake by masking β-galactosides on the surface of exosomes. Finally, galectin-9 in the lumen of exosomes released by nasopharyngeal carcinoma cells can induce inhibitory immunomodulatory effects on restricted populations of T-cells [65,66]. Although all of these galectins have been found in purified exosomal fractions, the question is where they are exactly localized. Some of them were described to be in the exosome lumen and some on their surface. This spatial information is critical to assess the biological function of the individual galectin in exosomal cell-to-cell communication. Certainly, the exact positioning of an exosome component depends on the cellular recruitment machinery.

## 5. Non-Classical Secretion of Galectins

Galectins are synthesized on free ribosomes in the cytosol [67]. They exhibit no signal sequence and they have to traverse the membrane by non-classical secretion mechanisms.

A possible mechanism for galectin export from the cytosol to the extracellular matrix has been described for fusion proteins of galectin-3 in COS cells [44]. The galectin-3 chimera, which was used in these experiments, is directed to the cytoplasmic sheet of the plasma membrane via an attached Lck-segment. These studies could also demonstrate that the N-terminal domain of galectin-3 is sufficient to direct membrane translocation. Both of these aspects are noteworthy since recruitment to the membrane and an at that time unknown essential secretion signal in the N-terminus of galectin-3 (see below) are reminiscent of the secretion mechanism of ARRDC1, which pinches off from the plasma membrane using the endosomal sorting complex required for transport (ESCRT) [68]. Strikingly, this process resembles the release of enveloped viruses on the plasma membrane, which use a so-called late domain PS/TAP motif to recruit the cellular ESCRT-machinery and thus bud off the plasma membrane [69,70]. Viral proteins have such a PS/TAP amino acid sequence in order to directly bind the Ubiquitin E2 variant domain of the ESCRT subunit tsg101 and thereby start the Vps4-mediated secretion.

Given the numerous studies that have equally demonstrated the localization of galectin-3 in exosomes by proteomic, biochemical, and microscopic detection, an export mechanism for this lectin using exosomal secretion appears very likely. In line with this, we were able to show recently that galectin-3 is packaged and secreted in exosomes, both topologically and mechanistically very similar to ARRDC1 and the secretion of enveloped viruses, through a PSAP-mediated, direct interaction with Tsg101 [71] (Figure 2). Interestingly, the galectin-3 tetrapeptide, which acts practically as a late-domain-like motif, is located in the N-terminus and is highly conserved. Incidentally, this raises the question of whether galectin-3 was an intrinsic template for the molecular process of viral release patterns [72,73]. Importantly, the application of specific inhibitors, the knockdown of tsg101 and the expression of the dominant-negative Vps4aE228Q mutant not only prevented exosomal secretion of galectin-3, but also the release of extracellular galectin-3 into the cell culture medium in general. This leads to the cautious conclusion that the transport route via multivesicular bodies (MVBs), intraluminal vesicles (ILVs), and finally exosomes could represent the exclusive secretion pathway for extracellular galectin-3, at least in epithelial Madin-Darby canine kidney cells. In other words, does the entire pool of extracellular galectin-3 result from the exosomal fraction? This question should be clarified in future work. Interestingly, the galectin-3 content of cell culture medium [71] or blood [51] correlate with the exosomal galectin-3 fraction, which indicates that nascent galectin-3 originated from the exosomal pool. This also raises the question of how galectin-3 can be released from the lumen of the membrane enclosed exosomes. Unfortunately, there is very little work in this area so far, so future studies will be needed to answer these questions.

In particular, the precise mechanism of galectin translocation across the membrane bilayer remains poorly understood. According to data published by Lukyanov et al., galectin-3 can spontaneously penetrate the lipid bilayer of liposomes by direct interaction with membrane lipids [74]. A possibly similar mechanism has also been described for microvesicular galectin-3, which could also cross the membrane in situ [44]. Direct translocation of galectin-1 on the other hand requires molecular machinery composed of integral and peripheral membrane proteins [75]. In fact, in the baker’s yeast *S. cerevisiae* exogenously expressed galectin-1 was secreted by non-classical secretion in dependence of the multidrug resistance homologue ABC transporter Ste6p [76]. Further mechanisms for galectin-3 release from secreted exosomes are conceivable, which are based on exosome-internalization and their subsequent degradation. Lysosomes, secretory lysosomes or even lysosomal exocytosis could play a role here [77]. By interacting with different lysosome-associated membrane glycoproteins [78,79], galectin-3 could be effectively protected against proteases. In fact, galectin-3 was detected by proteomic analyses in secretory lysosomes and melanosomes [80,81]. Another possible release mechanism could be based on secreted lipases, e.g., in an inflammatory microenvironment. This would ensure that certain exosomal proteins are released at their site of action. Interestingly, such a process has already been discussed for IL-1β [82]. In summary, fundamental studies are still required to evaluate the mechanism and the physiological relevance of the release of galectin-3 from exosomes.

That also applies to other still unknown aspects of the sorting process on the multivesicular bodies. In this context, future studies should clarify all components that interact in galectin-3-recruitment to the MVB. In our work there is evidence that galectin-3 is not only exclusively recruited to the MVB membrane via PSAP-UEV interaction with tsg101, but by further interaction partners including Alix (see also [83,84]). This coincides with the observation by Nabhan et al., in which an ARRDC1 mutant with a modified late-domain-like motif was still directed to the plasma membrane but no longer secreted [68].

There are other aspects of this interaction between galectin-3 and tsg101 in multivesicular bodies that are of great scientific interest. This includes, for example, the SNP (dbSNP:rs4652; PSAP→PSAT) in the human PSAP sequence. A PSAT sequence is very often found in the Eurasian part of the human population. How does this SNP affect exosomal secretion and interaction with tsg101, and what does this mean for the serum level of galectin-3 [85]? Are there certain diseases associated with the SNP? [86]. Previous scientific work in this area unfortunately does not clarify this point [87,88]. Thus, future work should definitely shed more light on this aspect. In this context, it would be very desirable to be able to receive crystallographic data of the galectin-3/tsg101 interaction in order to assess the influence of amino acid exchanges due to this SNP. Elucidation of the structure would also, incidentally, enable completely novel crystallographic insights into the 3-dimensional arrangement of the N-terminus of galectin-3, which has yet not been described. The same applies to the ASAA mutation of the binding motif, which has often been used in the viral context to prevent recruitment of the ESCRT complex [89,90]. It would be very interesting to see how this mutation alters the structure of the N-terminus of galectin-3.

But how do the other members of the galectin family, for which exosomal secretion has been described, get into exosomes? The highly conserved PSAP domain seems to be reserved exclusively for galectin-3. However, it would be very conceivable that galectin-3, by interacting with other galectins, could serve as an adapter for their packaging into exosomes. Such a mechanism would thus be a direct analogy to the exosomal secretion of syndecan and the lysyl-tRNA synthetase [91,92]. From a mechanistic point of view, the sorting process by syntenin exhibits several parallels to PSAP-mediated galectin-3 secretion. Syntenin interacts directly with Alix via three (L)YPXnL-motifs and thus imitates the second class of late domains that have been described in viral budding [93] in order to get secreted by exosomes. With the help of the PDZ-domains, Syntenin acts as a link between membrane and syndecan, as well as with Alix [91,94]. This leads to complexes that induce the pinching off and accumulation of ILVs. What can be derived from this for the other galectin representatives? Interestingly, galectin-9 (LYPsksiLL) and galectin-5 (YPnL) also contain amino acid sequences that could possibly act as an interaction motif for Alix. In addition, galectin-16 has a PPsY motif, which is also known from the viral context as a third type of late domain and which couples to NEDD4 family E3 ubiquitin ligases. In the future it will be extremely interesting to examine these putative late domains of other galectins and determine their importance for their incorporation into exosomes.

## 6. Conclusions

The inherent ability of galectins to leave the cytosol as their place of synthesis and enter the lumen of vesicular organelles is vital to ensure cell protection against lysosomal damage, subcellular cargo sorting and galectin-release for cell-to-cell communication at the organismal level. Currently, spatial and temporal aspects of galectin tasks on vesicular membranes have to be clarified in detail. Which are the interaction partners and how are these interactions dynamically controlled? Structural details will also aid to fully understand the sequence of events that lead to galectin functions. Expression and intracellular localization of galectins as well as their release into extracellular vesicles can provide a diagnostic and prognostic biomarker for several disease states. These lectins serve fundamental roles in the pathogenesis of human diseases, especially in the progression of cancer, fibrosis, and chronic inflammation [95]. The task for future therapeutic strategies will thus be to develop ingredients that specifically manipulate the function of individual galectins as cell sensors or sorting receptors and to control their incorporation into vesicles released by the cell.

## Figures and Tables

**Figure 1 biomolecules-10-01232-f001:**
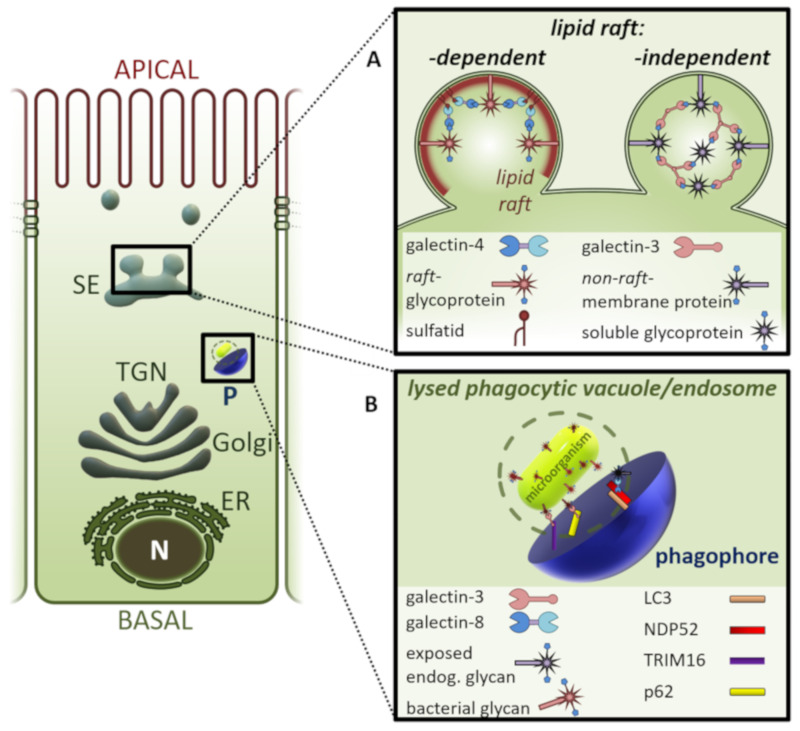
Galectins sort glycoproteins destined for the apical membrane domain of epithelial cells (**A**) and recruit the autophagy machinery for autophagosome formation (**B**). (**A**) Clustering of lipid raft-associated glycoproteins by galectin-4 or non-raft-glycoproteins by galectin-3 in the lumen of post-Golgi sorting endosomes directs apical cargo molecules into apical transport vesicles. (**B**) Galectin-mediated glycan detection on damaged endosomes or lysed phagocytic vacuoles recruits the autophagy receptors NDP52, TRIM16, or p62 to initiate nucleation of the phagophore in the cytosol. A phagophore then expands by lipid-acquisition to generate the sealed autophagosome. ER, endoplasmic reticulum; LC3, microtubule-associated protein 1A/1B-light chain 3; N, nucleus; NDP52, nuclear dot protein 52 kDa; P, phagophore; SE; p62, receptor of autophagy; sorting endosome; TGN, trans Golgi network; TRIM16, tripartite motif containing 16.

**Figure 2 biomolecules-10-01232-f002:**
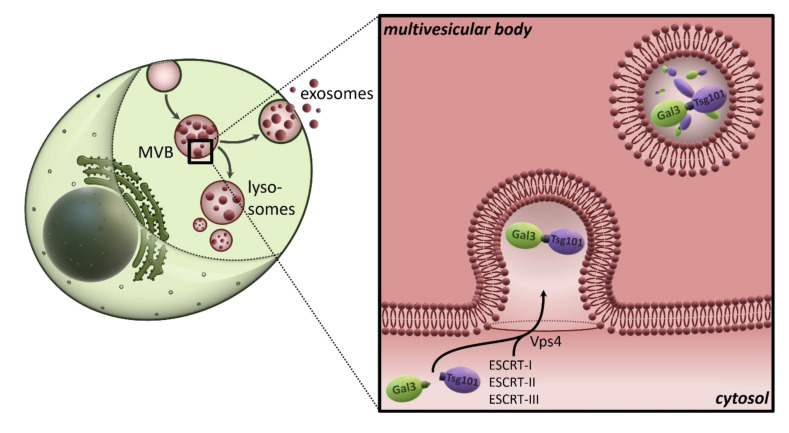
Exosomal secretion of galectin-3. A conserved P(S/T) AP motif in the amino-terminal region of galectin-3 directly interacts with the endosomal sorting complex required for transport (ESCRT)--component Tsg101. This interaction recruits galectin-3 into newly formed intraluminal vesicles (ILVs) of the multivesicular body (MVB). ILVs are sorted to lysosomes for degradation or released as exosomes into the outer milieu.

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
