# Peer review of "Galectins in Intra- and Extracellular Vesicles"

_biomolecules, 2020, doi:10.3390/biom10091232_

Round 1
Reviewer 1 Report
The review summarizes the current knowledge about galectin’s presence in intra- and extracellular vesicles.
The manuscript is well written and the summarized current state of the art is satisfying, even if the numbers of references are not written, so it is impossible to easily follow the choice of references made by authors. Please correct that.
I have found the review easy to follow and with an interesting new subject. Anyway, in my opinion, authors should emphasize what they state at the end of conclusion. They suggest that the knowledge of galectins and their role in vesicles could be useful for therapeutic purpose. Authors could justify this statement. The review has a substantial molecular intention and the clinical aspect has never been addressed. So I think that review could be improved by explaining the clinical topic that authors just mentioned.
Figures have really a poor resolution, images and written are blurry.
I suggest to revise the English because I have by myself found some mistakes:
Page 2 line 51, change “were” with “where”
Page 4 line 140, change “provide” with “provides”
Author Response
We wish to thank the reviewers for their positive comments and constructive criticisms. The manuscript has been revised according to their suggestions.
In the revised version, we emphasize the role of galectins played in different diseases and conclude that future therapeutic strategies will depend on the use of ingredients to manipulate these functions.
The revised manuscript has now been corrected by a native english speaking colleague and figure quality has been improved.
References are now indicated correctly in the revised manuscript.
Reviewer 2 Report
This is a short, informative review on an important topic. The only problem I have with this manuscript is that the text assigns a number to each citation, yet the references at the end of the manuscript are not listed by number. This needs to be consistent for readers to readily find the references cited in the text.
Author Response
We wish to thank the reviewers for their positive comments and constructive criticisms. The manuscript has been revised according to their suggestions.
References are now indicated correctly in the revised manuscript.